# A Deep-Learning Approach to Soil Moisture Estimation with GNSS-R

**Thomas Maximillian Roberts [1,2,\*]**, **Ian Colwell [2]**, **Clara Chew [3]**, **Stephen Lowe [2]** and **Rashmi Shah [2]**

1  Muon Space, Mountain View, CA 94043, USA
2  Jet Propulsion Laboratory, California Institute of Technology, Pasadena, CA 91109, USA;
   ian.colwell@jpl.nasa.gov (I.C.); steve.lowe@jpl.nasa.gov (S.L.); rashmi.shah@jpl.nasa.gov (R.S.)
3  COSMIC, University Corporation for Atmospheric Research, Boulder, CO 80307-3000, USA; clarac@ucar.edu
\*  Correspondence: max@muonspace.com

**Abstract:** GNSS reflection measurements in the form of delay-Doppler maps (DDM) can be used to complement soil measurements from the SMAP Mission, which has a revisit rate too slow for some hydrological/meteorological studies. The standard approach, which only considers the peak value of the DDM, is subject to a significant amount of uncertainty due to the fact that the peak value of the DDM is not only affected by soil moisture, but also complex topography, inundation, and overlying vegetation. We hypothesize that information from the entire 2D DDM could help decrease uncertainty under various conditions. The application of deep-learning-based techniques has the potential to extract additional information from the entire DDM, while simultaneously allowing for the incorporation of additional contextual information from external datasets. This work explored the data-driven approach of convolutional neural networks (CNNs) to determine complex relationships between the reflection measurement and surface parameters, providing the groundwork for a mechanism to achieve improved global soil moisture estimates. A CNN was trained on CYGNSS DDMs and contextual ancillary datasets as inputs, with aligned SMAP soil moisture values as the targets. Data were aggregated into training sets, and a CNN was developed to process them. Predictions from the CNN were studied using an unbiased subset of samples, showing strong correlation with the SMAP target values. With this network, a soil moisture product was generated using DDMs from 2017–2019 which is generally comparable to existing global soil moisture products, and shows potential advantages in spatial resolution and coverage over regions where SMAP does not perform well. Comparisons with in-situ measurements demonstrate the correlation between the network predictions and ground truth with high temporal resolution.

**Keywords:** GNSS-R; CYGNSS; SMAP; soil moisture; deep learning; convolutional neural network

## 1. Introduction

Accurate and timely quantification of near-surface soil moisture is important for many hydrologic and atmospheric applications. The upper few cm of the soil constitutes the boundary between the land surface and the atmosphere, and the relative partitioning between sensible and latent heat fluxes is governed by the moisture content of the soil [1]. The evaporation of water from the soil back into the atmosphere can influence subsequent precipitation events, thus making knowledge of soil moisture important for numerical weather forecasting [2]. In addition to the soil moisture–rainfall feedback, observation of near-surface soil moisture can also serve as both early indications of drought as well as flood risk [3]. For over two decades, remote sensing of soil moisture has been further enabled by the development of methods that leverage reflected signals from Global Navigation Satellite Systems (GNSS), which are affected by the soil moisture content of the reflecting surface. Signals collected from tower-mounted antenna have been demonstrated to accurately measure the soil moisture content using various interferometeric and power-based techniques [4–7]. Measurements have also been made from aircraft, allowing for the

testing of novel instrumentation and techniques over wider areas [8–12]. The collection of reflected GNSS signals from orbital platforms provides a critical global perspective with unparalleled coverage, and will be the focus of this work.

Passive microwave measurements from the Soil Moisture Active Passive (SMAP) Mission [13] are the current standard for global soil moisture estimation from a remote platform, but with a revisit rate of 2–3 days, many climate/hydrological studies would benefit from improved spatiotemporal resolution [14]. Similarly, weather/flood prediction needs soil moisture measurements with spatial resolution better than 10 km, and revisit times of less than 3 days [15], a requirement the Cyclone Global Navigation Satellite System (CYGNSS) constellation [16] of eight spacecrafts can achieve using the reflected signals from GPS transmitters. Figure 1 shows a visual comparison of the daily coverage (top) and attainable spatial resolution (bottom) between SMAP and CYGNSS. Additionally, as there are no current plans for another SMAP-class mission, and constellations, such as CYGNSS, are an order of magnitude less expensive to build and operate, Global Navigation Satellite System reflectometry (GNSS-R) is likely to be a dominant contributor to future remote soil moisture estimation [17–20]. A soil moisture-calibration concept has been demonstrated in work by Chew [21], where a linear relation between the peak value of reflected GPS power as measured by CYGNSS and soil moisture as measured by SMAP is assumed. While broadly successful, this simple technique can fail in regions where there is low variation in soil moisture or where confounding factors (vegetation, surface roughness, topography) complicate the relation between power and soil moisture.

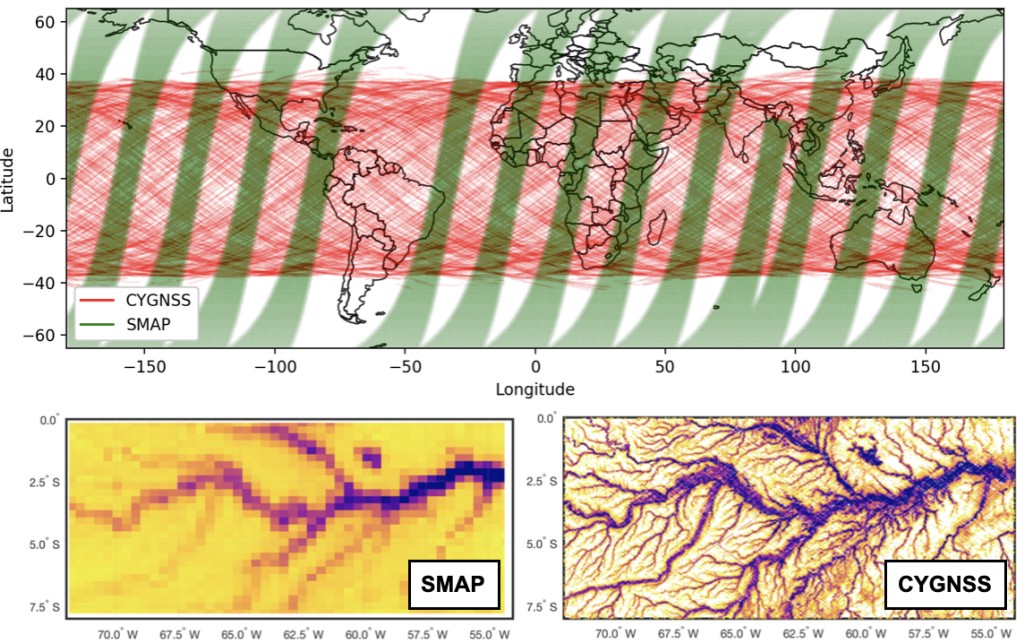

**Figure 1.** Comparison of SMAP and CYGNSS daily coverage (**top**) and attainable spatial resolutions (**bottom**).

The GNSS-R measurements made by CYGNSS take the form of delay-Doppler maps (DDMs). These 2D arrays appear markedly different over Earth surfaces with varying conditions (as depicted in the small panels on the left of Figure 2) but are analytically challenging to quantify. Traditional analyses of DDMs often only utilize a single profile (1D slice, right of Figure 2), or, as in the work of Chew, a single value, resulting in 90–99% of the data contained in these measurements being discarded due to the complexity of full interpretation. The application of deep-learning-based image-interpretation techniques to the analysis of DDMs has the potential to extract information from the entire 2D DDM, while simultaneously providing the option to incorporate additional contextual information from external datasets. This approach has advantages over existing machine-learning-based

analyses [22–24], which still rely on single-valued feature extraction from the DDMs, and has been previously used in the classification of sea ice [25].

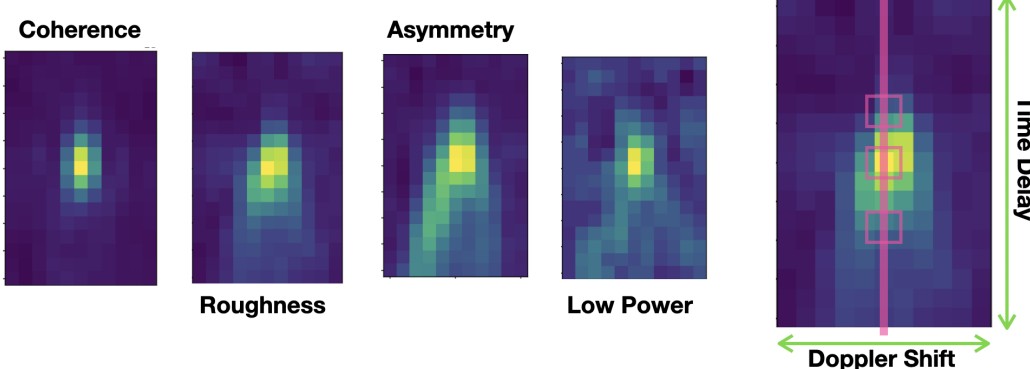

**Figure 2.** Examples of DDM features with changing surface conditions (1–4, from left to right). An example of data reduction through selection of a single profile or individual points (rightmost, extracted values represented by line and squares).

The work described in this article details a preliminary exploration of the use of convolutional neural networks (CNNs) to learn these relationships and provide a mechanism to achieve improved soil moisture measurements, especially in regions where simple linear relations yield high uncertainty or fail. By inputting values without constraints imposed by simplifying assumptions about which features are important, data are allowed to drive the interpretation and the network effectively creates a model that determines the complex relations between the dataset values. Figure 3 presents an overview of the data inputs, targets, and output of the developed CNN. CYGNSS DDMs acted as the primary input values and are trained with SMAP soil moisture measurements as targets. Values from ancillary datasets were included as additional inputs to provide context for the reflection measurement. As will be detailed in later sections, the network was trained, validated, and scored on separate partitions of the data to ensure performance metrics are representative of the network's true ability to estimate soil moisture from new/unseen DDMs. The development of the CNN-based analysis pipeline was an iterative process where the datasets were built, the neural networks trained, and results studied, repeatedly. This allowed for the rapid detection of issues with the datasets, and testing of improved neural network implementations.

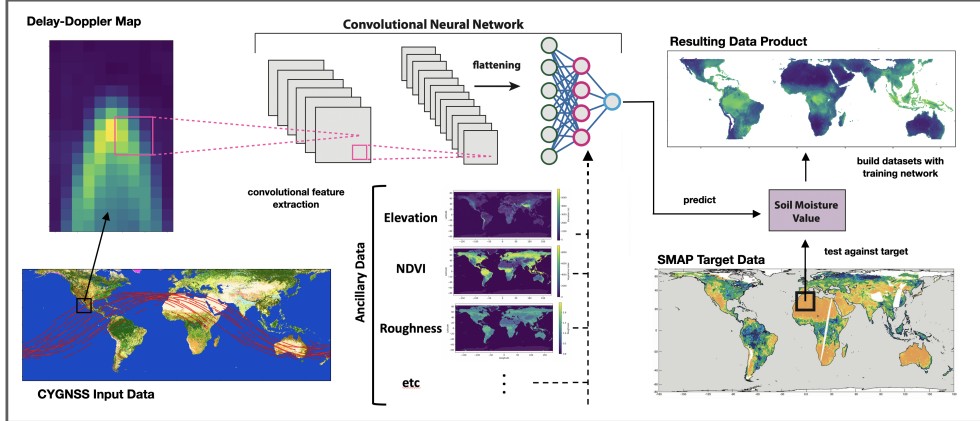

**Figure 3.** Overview of surface parameter estimation with a convolutional neural network using delay-Doppler maps and ancillary data as inputs, and SMAP soil moisture values as targets.

The following sections review three major tasks: (1) dataset preparation, (2) neural network development and training, and (3) prediction evaluation. The results of the trained

network are then discussed and compared to existing global soil moisture products, and we conclude with areas for further enhancement and alternative applications.

## 2. Materials and Methods

This section is divided into two major subsections. First we focus on the dataset development, including selection of the utilized datasets with detailed descriptions, followed by discussions of data alignment, filtering, and partitioning methods. Next, we look at the development of the neural network, which is broken into two separate parts; a brief study of the DDM-specific sub-network is presented, followed by a description of the techniques used to build the complete CNN for soil moisture.

### 2.1. Dataset Preparation

Dataset preparation involved several key steps. First, we determined which datasets would be used as inputs and targets. These data were then studied to understand their nature and distributions. With this knowledge, filtering was applied to ensure reasonable behavior. With the data properly down-selected, samples appropriate for ingestion by the neural network were created.

#### 2.1.1. Selected Datasets

This subsection describes important details about the datasets and comments on preparation for input to the training set. Note that all the datasets vary by location, but some are assumed static for the duration of time considered by this study (e.g., topography) while others vary daily or are specific to individual DDMs. All of the datasets were preprocessed from their original formats into HDF5 format files for ease of integration. In order to compare the results of this work to Chew's global soil moisture product [26], we chose to use SMAP and CYGNSS data of the same product levels and resolutions as used in the UCAR soil moisture product. Training was performed using the same calibration window (18 March 2017 to 1 October 2018), while product comparisons were formed over the matching validation window (2 October 2018 to 31 December 2019). Continuous, quantitative values, such as the DDM input, soil-moisture output, and certain ancillary datasets were standardized to zero mean and unit variance prior to input to the network to ensure numerical stability.

#### 2.1.2. Delay-Doppler Maps

The primary input is the CYGNSS L1a [27] "power_analog" DDM matrix, a $17 \times 11$ element array of calibrated power (in Watts) from a reflecting surface where each array value represents the power at a specific time delay and Doppler shift. Each CYGNSS spacecraft produces simultaneous DDMs of reflections from 4 different transmitters at a 1 Hz rate. Neglecting outages and calibration windows, the entire constellation produces on the order of $10^9$ measurements per year. The power contributing to a particular DDM can come from a region as wide as 25 km if the surface is rough (incoherent scattering), or as small as a sub-kilometer body of water if this surface is smooth (coherent scattering). As such, describing the spatial resolution of a particular reflection can be challenging. Over many land surfaces, we can consider the coherent part of the signal to dominate, and, therefore, assume a minimum footprint of $0.5 \times 7$ km, elongated along-track due to the receiver motion.

The pixel values range roughly from $10^{-16}$–$10^{-18}$ Watts, and therefore must be normalized for numeric stability. This was performed by calculating the mean and standard derivation for over the calibration window, and standardizing by these values to achieve zero mean/unit variance. It was observed that the DDM amplitude distribution varied slightly between the spacecraft; therefore, DDMs were normalized using means/variances as calculated for that spacecraft. These normalization terms were calculated after problematic values had been removed from the dataset.

The CYGNSS product includes a quality flag parameter that indicates measurements with potentially erroneous features. Certain flags are used to definitively remove data that is of no interest to this study, such as measurements made over the ocean or during instrument calibration, though others need to be considered more carefully. An example is the RFI flag, which indicates the possibility of radio interference during the measurement window. These flags are so common over many regions of the continental US and Europe that, if used for exclusion, they would substantially limit the coverage of our process. As such, this flag was ignored. A separate technique was developed to flag significant RFI in DDMs, by looking for high variances in portions of the DDM devoid of reflected signal, indicating the presence of interference sources.

### 2.1.3. Spacecraft Ancillary Data

In addition to the DDMs just discussed, every CYGNSS L1 [27] measurement includes an array of information about the spacecraft configurations, geometries, and antenna parameters that play an important role in the reflection characteristics. As the power collected by the antenna is a function of the receiver/transmitter separation, the transmitter power, and the receiver antenna, these values are included as ancillary input for every sample. Additionally, we used these parameters to explore a modification to the 2D DDM input suggested by Chew [21], which corrects for these inputs ahead of time. The model relates the reflected power, $P_r$, to an effective surface reflectivity, $\Gamma$, as

$$P_r = \frac{P^t G^t}{4\pi(R_{ts} + R_{sr})^2} \frac{G^r \lambda^2}{4\pi} \Gamma \tag{1}$$

where $P$ is power, $G$ is gain, $R$ is separation and the $t, r, s$ subscripts represent the transmitter, receiver, and specular reflection point, respectively. $\lambda$ is the wavelength of the GPS L1 signal, 19.05 cm. Note that this expression is for a coherent reflection, which is not a correct assumption for many elements of the 2D DDM. While this approach does effectively impose a model on the reflection, this model is well-defined and decouples surface effects from range/antenna effects, creating a 2D $\Gamma$ input as a replacement for the raw DDM input. As such, four configurations of inputs were explored: (1) DDMs alone, (2) DDMs with range/antenna ancillary inputs, (3) $\Gamma$ alone, and (4) $\Gamma$ with range/antenna ancillary inputs. The fourth permutation yielded the fastest training times and smallest training residuals. As such, all DDMs were preprocessed in this manner. Development of a more appropriate treatment to the input DDMs will be part of future work, but was out of scope for this study.

Distortions to the DDM can be created by large reflection incidence angles, which are not accounted for in the model mentioned above. As such, the azimuth and elevation angles of the reflection location, relative to the receiver, are important in this context, but also for filtering purposes, discussed later in this section. These angles are, therefore, included as inputs.

The location (latitude and longitude) of the measurements is also an important input to include. This allows the network to learn regionally specific soil moisture behaviors, analogous to the empirical relations formed for the generation of other soil moisture products [26]. This is one of the strongest ancillary inputs, allowing for reasonable estimation in disparate regions of the planet (Sahara vs the Amazon) but is still heavily reliant on the primary and other ancillary inputs to perform well due to temporal/seasonal variation of soil moisture at a fixed location. The treatment of angular values can represent an interesting challenge where locations are multi-valued ($\pm 180°$ longitude) or poorly behaved ($\pm 90°$ latitude). As these locations are of little interest for this work, we simply input the direct angular values with no modification.

Two categorical parameters of interest are the spacecraft and transmitter ID values. These are "label"-type data where numbers hold no meaningful relation to one another. As such, we used a unique-value binary representation, "one-hot" arrays, to map the scalar inputs to vector inputs of length equal to the number of unique value possibilities. Using the example of the 8 CYGNSS spacecraft, a value of "2" would be mapped to an 8-element

array where all values are zero except the second element, which is 1. This technique was also implemented for land type, as discussed in the next subsection.

Finally, DDM features, similar to the features used in traditional analyses of DDMs, were extracted and input in isolation. This was performed for the DDM and $\Gamma$ peak values, allowing not only for studying the impact of including this supplemental data, but also enabling us to observe performance when using these peaks as a replacement input for DDMs. Tests of this nature allow us to represent inputs more similar to the linear analysis of Chew and further evaluate the benefits of a CNN-based approach.

### 2.1.4. Surface Ancillary Data

Previous studies and discussions with experts indicated that key surface factors influencing the reflected signal power include the surface topography, roughness, surface water, and vegetation. Several datasets were aggregated to represent these parameters, and selection of these datasets was performed through observation of the impact of inclusion of the datasets on reduction in the training residual and agreement between the prediction and target distributions. A simple averaging of the dataset values within a 3 km radius of the specular reflection point provided the input for a particular parameter. For the case of land type, only the value of the maximum fractional type was used.

Surface topography was represented by the SRTM digital elevation maps via the SMAP ancillary dataset [28], which include both terrain elevation and slope. Elevation is expected to play little role in the DDM form as the delay has been adjusted to account for surface height, but is useful to filter DDMs that occur from high regions where CYGNSS is known to have performance issues, and very steep surfaces which may or may not reflect the signal towards the receiver. Surface roughness data were also pulled from the SMAP ancillary datasets. While roughness is an important parameter for both CYGNSS and SMAP, as their respective sensitivity wavelengths differ, so does what is considered "roughness". While this parameter was found to yield little impact on the training process, a roughness parameter appropriate for CYGNSS is under development at JPL, and will be incorporated in future analyses.

Surface water is a strong contributor to the structure and amplitude of a reflected signal, and so received additional attention. The Pekel water mask [29] is a community standard for statistical representation of surface water. Surface water occurrence, the likelihood of surface water at a location, was used as the input parameter. This value can have a seasonal variability at a given location, but for this study we used a fixed annual value. The water masks are high resolution (30 m), so a preliminary step was to downscale the resolution to 3 km grid cells by two-dimensional averaging.

The presence of vegetation has a significant impact on the emissions and attenuation of wavelengths measured by SMAP [30,31], as well as scattering/attenuation effects for CYGNSS. This is primarily due to the water content in the leaves, stems, branches, and trunks of plants. A parameter representing the vegetation water content (VWC) was also supplied as an input, which is calculated using a empirical parameter known as "stem factor" in combination with the normalized difference vegetation index (NDVI) [32]. NDVI is available as a daily parameter in the SMAP ancillary dataset [28], while the stem factor is static. While we chose to use both NDVI and VWC as inputs, VWC was found to have a stronger influence on training. Future work to determine individual feature contributions to the network's performance will highlight if providing both NDVI and VWC separately provides additional benefits over using just one of these features as input to the network.

The only categorical surface data used was land type, provided by the GlobCover [33] dataset. Here, the most common value (highest fractional content) found in the 3 km area around the reflection was used as input. These values were then mapped to "one-hot" array (detailed in the previous section). This simple approach will be improved upon in future datasets by instead inputting a land-type vector with fractional land-type values for each input.

While many of the above ancillary inputs were found to be strong predictors of soil moisture during training (such as location, VWC, and surface water occurrence), others yielded little discernible improvement in the network's performance. In general, if an ancillary value ends up being uncorrelated with the target variable, the network will learn to ignore it as an input. Determination of an optimized configuration of these inputs was out of the project scope, so all the inputs discussed above were used. A future study will explore the predictive contributions of these individual datasets in detail.

### 2.1.5. Soil Moisture

SMAP soil moisture values were used as the single-valued metric the network aims to predict, or the target, and have units of $m^3/m^3$ representing the volume of water per volume. The SMAP L3 product [34] bins all measurements from a single day to a 36 km EASE [35] grid. These are given as "AM" and "PM" values for the different local times that SMAP passes a region. While the SMAP mission requirements for soil moisture measurements of 0.04 $m^3/m^3$ unbiased RMS error [13] (ubRMSE) are attained, both AM and PM SMAP soil moisture values suffer from spatially and temporally varying biases [36]. It is expected that PM soil moisture values will suffer increased variance and bias due to thermal gradients in soil and canopies [37]; therefore, only the AM values were used for this work. It should also be noted that the lowest reported SMAP value is 0.02 $m^3/m^3$, yielding a distinct cutoff in the target distribution.

A typical day of AM values is shown in green in the upper plot of Figure 1. Note that the daily coverage for the latitudes where CYGNSS operates is on the order of 30–50%. As such, there are many CYGNSS measurements for which there are no corresponding SMAP measurements.

SMAP, like CYGNSS, provides quality flags for these measurements in the form of a matching daily EASE grid. These flags indicate whether a soil moisture measurement was successful, completely failed, or was somewhat unreliable. Figure 4 shows the distributions of these flags with soil moisture for the entire year of 2018. Note the majority of soil moisture measurements above 0.5 $m^3/m^3$ are marked as problematic in one form or another. For this work, measurements associated with the blue, orange, green, and red distributions (first four in legend) were included, excluding the black and purple. This selection was a compromise between measurement reliability and sufficient global coverage, as many regions were entirely composed of "not recommended" soil moisture values.

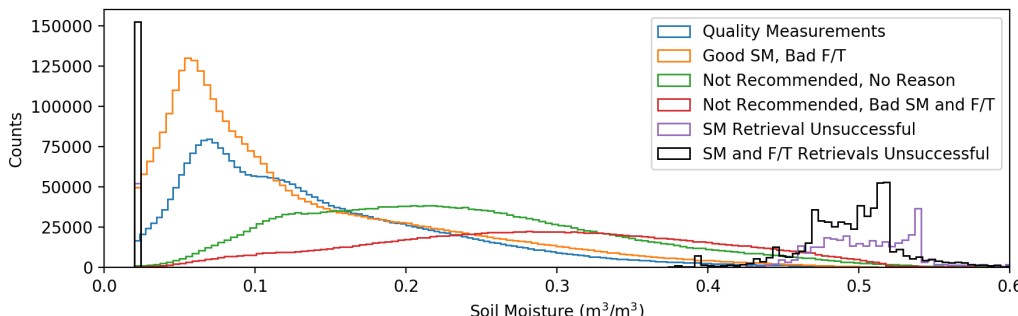

**Figure 4.** SMAP quality flag distributions with soil moisture. Large spike at low values from black/purple distributions.

### 2.1.6. Data Alignment

Having defined the datasets of interest, the spatiotemporal alignment of these various parameters was the next step in building a training database. For each CYGNSS DDM, the corresponding SMAP soil moisture value and contextual information needed to be matched to create a "sample" used for training or testing of the neural network. Inherently, this necessitated accommodation of differences in spatial resolution and timescales to choose the "nearest" measurement.

As detailed in the previous section, all ancillary datasets were mapped to 3 km resolution EASE grid cells (if not provided at this scale), while the original resolution of the SMAP L3 soil moisture data of 36 km was retained for target values. This was, in part, an effort to best match the approach taken by Chew, while providing a final product at a finer spatial resolution. Temporal alignment was achieved with the simplifying assumption of "same day" values. Therefore, the UTC day that the DDM occurred on was paired to the SMAP value of the same day. The same approach was used for temporally varying ancillary datasets. Improvements in the spatiotemporal alignment/resolution are left for future work, but would include 2D interpolation between grid cells to better approximate the local distributions, as well as interpolation in time to fit the nearest SMAP (and time-varying ancillary) values.

By performing this alignment for every valid CYGNSS measurement with a corresponding SMAP value in the calibration window, a sample database was constructed. As detailed in the next section, this database could be parsed under various requirements to create the training data for the network. Separate from this database, a significant number of CYGNSS DDMs exist that could not be aligned to SMAP measurements during the calibration window. These values form a separate input-only database where DDMs are aligned to all the necessary ancillary data to be fed into a trained network, effectively extending the validation window. These data are not focused on in subsequent discussion though; instead, we only consider the calibration window of late 2018–2019.

### 2.1.7. Filtering

Filtering of these datasets was a critical preprocessing step, and one of the key tools for adjustment in the iterative development process. While filtering has been discussed in some of the above sections, here we define additional criteria for retaining samples in the training set for the discussed parameters. Filtering is performed prior to training simply by discarding samples from the complete database that fail to meet the filtering criteria. Therefore, for a given training session, different filtering criteria could be set, and training would only be performed with the remaining samples.

Filtering was divided into three dominant categories; filtering to data that could be aligned, filtering of "bad" data, and filtering of data to simplify training. The initial down selection of samples was based on CYGNSS quality flags, outlined in Section 2.1.1, which created the list of DDMs to align other data against. This was followed by only retaining samples that were successfully aligned with SMAP measurements that met the quality flags requirements detailed in Section 2.1.5. Following this, samples with ancillary data containing detected problematic values were removed (problematic defined through individual analysis of these sets). These remaining samples formed the training database, which could be dynamically filtered at training time.

Dynamic filtering of the samples was performed with conditional statements applied on certain ancillary data. Some filtering was applied based on expert input on the reliability of measurements, while other filters were used to limit the nature of the training to a specified parameter space. Surface elevation data were referenced to exclude DDMs that occur above 3 km elevation, as there are known issues with the reflection data at extreme heights. Locations with surface water occurrence greater than 1% were omitted to prevent inclusion of bi-modal surface states due to inundation. Limitations on acceptable angles from the spacecraft to the reflection point were imposed, allowing elevations of 20–60° (nadir is 0°) and azimuth angles of $\pm45°$ (90° is perpendicular to ram). This last constraint was an effort to simplify the parameter space by avoiding interpretation of the mostly heavily distorted and asymmetric DDMs.

Regarding the quality flags of CYGNSS and SMAP, a balance was stuck between filtering out "bad" values (non-sensible or likely incorrect) and achieving sufficient spatial coverage. Clearly, these choices are somewhat qualitative, and alternative approaches could be employed. Had we chosen to include only the most reliable samples as targets, the network would likely perform well in regions with measurements, and may struggle

in making reasonable predictions in unseen regions. Confirmation of the performance in these areas would present a challenge. Future work incorporating in-situ measurement of soil moisture could possibly address studies such as this, but for this work we chose to retain CYGNSS and SMAP flags with imperfect reliability to ensure coverage of South America and major regions of sub-Saharan Africa.

Depending on the specifics of the filtering applied, the entire CYGNSS constellation generates order 30–60 million training samples for late 2017–2018. For the results shown in Section 3.2, roughly 43 million training samples were retained during the calibration period, while the validation period had closer to 100 million samples for input to the trained network.

### 2.1.8. Sample Partitioning and Balancing

As is standard practice in neural network development, the filtered set of samples was split into training, validation, and test subsets that aided in the development and evaluation of the model. These sets were designed to provide sufficient data to train the network, evaluate its performance and aid in hyperparameter tuning, and provided an unbiased evaluation of how the network will perform when exposed to new unseen data. For this work, we chose to use an 80%/15%/5% split in an effort to include a sufficient amount of data in the test set to observe the extent of global coverage and uncertainty. The splits were randomly sampled to avoid including any unwanted artifacts related to how the data were created and stored, e.g., temporal-, spatial-, and spacecraft-related ordering. A random seed was used when generating the splits to ensure reproducible results.

The blue bars in Figure 5 show the distribution of SMAP target values in the sample database, effectively representing the global distribution of soil moisture. There is a clear dominance of samples from drier regions, creating a target distribution with a heavy bias toward low values. Training a neural network with a sample distribution such as this often results in a performance that is very accurate in predicting the common, dry targets, but has little exposure to the minority populations of high soil moisture values. As such, a network trained on this type of target distribution will severely underperform over wetter regions.

One solution to this issue is to create a training data subset that is "balanced" by over/undersampling to achieve an even target distribution, represented by the orange bars in Figure 5. Here, samples for a given soil moisture value are randomly selected until the threshold (orange line) is met, with repeated selection occurring for the less common, high soil moisture samples. In doing so, the network is evenly exposed to soil moisture of all values, helping to prevent bias in training. Note that this is performed for the training samples, not the validation or test samples. These subsets are retained with the original distributions to ensure they provide realistic measures of performance. Balancing was performed in a manner that preserved the same number of training samples as in the original distribution.

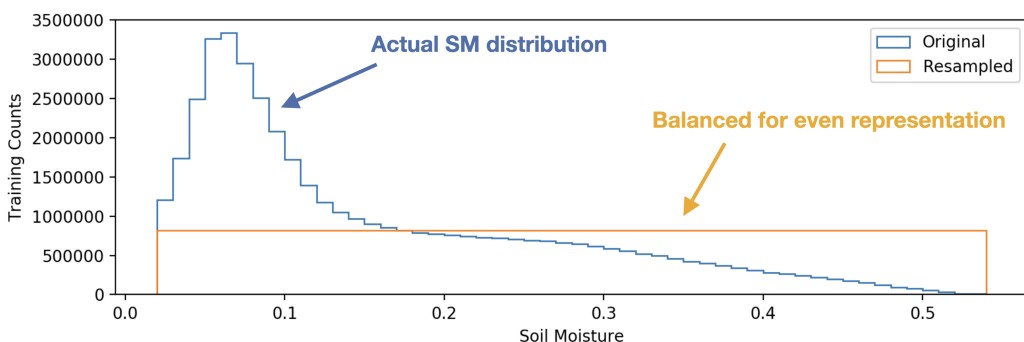

**Figure 5.** The original target distribution in blue compared to the balanced in orange.

While useful, this process does introduce two issues that must be considered. First, amplification of variability between repeated runs is driven by random selection of elements

in the training/validation splits. For high soil moisture values where there are relatively few samples, the samples that are split into these subsets will have a disproportionate influence on the estimates of performance depending on their relation to the samples that are repeated many times in the balanced training set. Second, due to the diminishing tail of the distribution at high target values, a reasonable cutoff must be set for the oversampling process. For this work, the cutoff usually fell between 0.5–0.55 m$^3$/m$^3$. As such, the training set does not include samples with soil moisture values above this level. This will inherently prevent agreement with the wettest measurements produced by SMAP, as will be discussed later on. Future work will explore improvements in this technique to ensure reliable, unbiased behavior.

## 2.2. Neural Network Development

The recent advances in the field of computer vision, seen in the development of new techniques and refinements to the use of CNNs in image-centric tasks, made the application of CNNs to DDMs particularly appealing. Conversations with subject-matter experts familiar with GNSS-R and DDMs would often involve the expert casually referring to a particular reflection as likely having occurred over land, water, a forested area, etc., based entirely on the structure of the DDM. These anecdotal experiences in conjunction with the image-like structure of DDMs provided sufficient motivation to develop a CNN capable of handling DDMs as inputs, while targeting a surface parameter.

Development of the neural network is inherently a coupled, nonlinear problem, but the subregions of the network can be somewhat studied in isolation. We split this task in two, developing a CNN that is specifically tuned for processing DDMs (blue layers of Figure 6), and developing a network architecture capable of accepting dynamic inputs for prediction of soil moisture, including the convolutional layers for processing DDMs (all layers in Figure 6). All of this work was performed in Python, mainly leveraging the TensorFlow and Keras libraries.

### 2.2.1. DDM-Tuned CNN Development

The CNN architecture and training process was developed with the "toy" problem of land-type classification; this provided the opportunity to independently develop strategies for improving a DDM-based CNN while the soil moisture pipeline was refined separately. Here, a DDM was fed as the sole input, and the target was a single value corresponding to a land cover index from the GlobCover 2009 dataset. Using this framework, several questions/techniques were explored: (1) do CNNs provide a benefit over an artificial neural network (ANN), (2) filtering of DDMs beyond quality flags, and (3) network architecture refinements.

The GlobCover 2009 data product covers 22 different land-cover classes, varying from vegetative areas, grasslands, bare areas, and water. As expected, the class distributions for land cover are highly imbalanced. In the context of exploring the applicability of CNNs to capture information from the complete DDM, we focused our analysis on two land-cover classes. The first class consisted of closed to open (>15%) broadleaved evergreen and/or semi-deciduous forest (>5 m) (GlobCover class ID-40) representing 8.17% of the GlobCover coverage. The second land-cover class used in this analysis corresponded to bare areas (GlobCover class ID-200), which covers 14.03% of the GlobCover data—the second largest class next to water bodies (14.34% coverage). These classes were selected as they are well sampled, ensuring there would be sufficient data to train a CNN, and represent distinctly different types of land coverage, providing a target that should be characteristically different enough for the CNN to distinguish between. When aligned with the 2017–2018 CYGNSS data, there were approximately 3 million observations between the 2 land cover classes. Training, validation, and test splits were created using a 80%/15%/5% strategy.

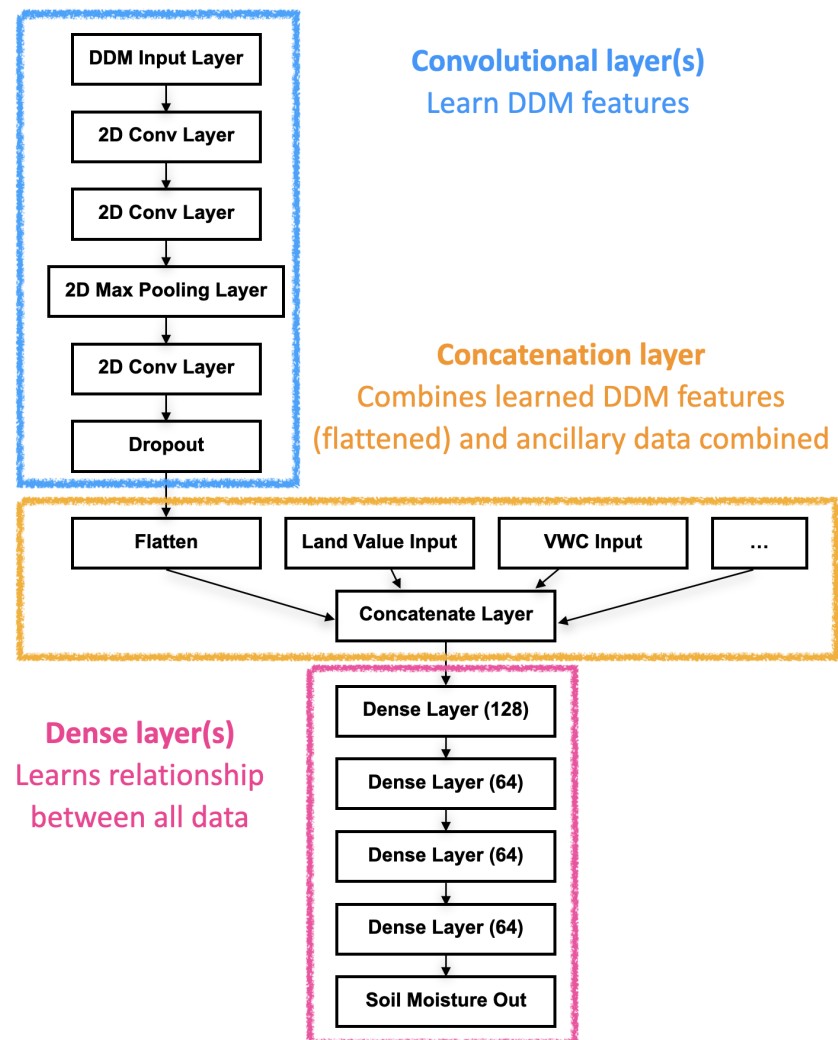

**Figure 6.** Layout of complete neural network (specific layers exemplary) highlighting functional regions. Here, 2D layers of the DDM subnetwork include convolutional layers, max-pooling layers which downsample using regional maxima from previous layer, and dropout layers which randomly remove inputs during training to prevent overfitting. The concatenation layer combines the output of the DDM layers with the ancillary data inputs, and passes these to several dense (fully connected) layers.

### 2.2.2. ANNs vs. CNNs

We compared the performance of an ANN and CNN trained on the full DDM to classify bare and forested areas; the same train, validation, and test splits were used across the different networks. The number of layers and trainable parameters were set to be consistent between the different architectures to ensure as even a comparison as possible. If the CNN under performed relative to the ANN, then it would reason that a CNN-based architecture may be ill-suited for DDMs as inputs. To accommodate the ANN, the 2D (17 × 11) DDMs were flatted into 1D vectors (length 187) prior to being input into the network; the full 2D DDMs were used as input to the CNN. Both network architectures had similar performance, accuracy around 88% and F1-score [38] of 88% (Figure 7). These performance metrics improved as the number of layers and trainable parameters increased across both architectures. The larger CNNs did see a slight improvement in performance over their ANN counterparts. As further confirmation of our hypothesis that using the full DDM would provide valuable contextual information, we compared results to a network trained using only the peak power as input (representing common methods). When only

the peak power was used as input, we saw the network's overall accuracy decrease to 75% and struggle with the class imbalance—F1-score was 75%.

| Architecture | Inputs | Detail | Layers | Parameters | Loss | F1-score | Accuracy |
|---|---|---|---|---|---|---|---|
| ANN | Peak power | | 7 | 10k | 0.51 | 75% | 75% |
| ANN | Flattened DDM | | 7 | 10k | 0.28 | 88% | 88% |
| ANN | Flattened DDM | | 8 | 200k | 0.25 | 89% | 89% |
| ANN | Flattened DDM | | 10 | 700k | 0.25 | 89% | 89% |
| CNN | DDM | | 7 | 10k | 0.27 | 88% | 88% |
| CNN | DDM | | 8 | 200k | 0.24 | 90% | 89% |
| CNN | DDM | | 10 | 700k | 0.23 | 90% | 90% |
| CNN | DDM | L2 regularization | 10 | 700k | 0.24 | 90% | 90% |
| CNN | DDM | Dropout | 10 | 700k | 0.23 | 90% | 90% |
| CNN | DDM | ResNet convolutional blocks | 10 | 700k | 0.25 | 89% | 89% |
| CNN | DDM | Augmentation applied to DDMs during training | 10 | 700k | 0.25 | 89% | 89% |

**Figure 7.** Performance comparison between the ANN with the peak power as input, ANN with the flattened DDM as input, and CNN with the original DDM as input. The networks presented used the same train, validation, and test splits, learning rate (0.0001), and number of epochs (10) to ensure a similar comparison while evaluating differences in inputs and changes in network architecture. The results corresponded directly to the test-data split (approximately 150k observations—5% of 3 m observations). F1-score is provided alongside accuracy due to the slight imbalance in the distribution of bare and forested land classes. The CNN with 700k parameters (highlighted in green) had the best performance in F1-score and accuracy, as well as the lowest loss.

### 2.2.3. DDM Filtering

After establishing confidence in a CNN-based architecture for this task, we considered different strategies to improve the performance of the network. With each iteration, we performed error analysis—examining DDMs that the model classified correctly with high confidence and, conversely, examining DDMs that the model classified incorrectly with high confidence. These misclassifications shed light on a series of structural issues with DDMs that were not being filtered by the CYGNSS quality flags. For example, as we were not excluding CYGNSS measurements with the RFI flag, we noticed many of the misclassified DDMs contained visible striations, indicative of significant RFI. In a similar manner, error analysis helped establish a series of additional quality filters that removed DDMs if: the mean of the DDM was below 0; the minimum value in a DDM was above 0; the maximum value of a DDM was found on the edge/border; or the maximum value was sufficiently large ($>10^{-13}$). Introducing these additional filters improved the training data and provided an enhancement in overall classification accuracy.

### 2.2.4. CNN Architecture Study

We examined how some structural changes to the network architecture impacted performance. During this process, we reviewed the advancements made by modern CNN architectures such as ResNet, VGG, AlexNet, etc., and borrowed their findings, to an extent. In the context of the land type CNN, we found that notions from AlexNet, such as dropout layers following the dense layers and image augmentation, improved model performance and generalizability. From VGG, we consistently utilized smaller kernels throughout the convolutional layers. Swapping out the vanilla convolutional layers for identity blocks and convolutional blocks, pioneered by ResNet, did not result in an immediate benefit to the CNN when restricting the number of trainable parameters to 700k.

We further employed image augmentation in the form of horizontal and vertical shifts in the range of $[-2, 2]$ pixels and reflecting about the vertical axis (Figure 8). At training time, each DDM was subjected to multiple random combinations of these augmentations while maintaining the same target (land type) value. Due to the process in which DDMs are constructed, performing a reflection about the horizontal axis would yield an unphysical interpretation of the DDM; as such, we intentionally did not include this form of image

augmentation. A range of different shifts in pixel values was experimented, but the range of $[-2, 2]$ was found to be optimal in the case of the original resolution ($17 \times 11$) DDMs. Coupling image augmentation with the largest CNN architecture did not provide an immediate benefit to performance. However, image augmentation combined with other regularization techniques, such as dropout, improved the network's overall performance (accuracy of 90% and F1-score of 90%) but at the cost of increased training times. An increase in the network's generalization, seen in the smoothing of the network's train/validation loss curves during training, was also observed.

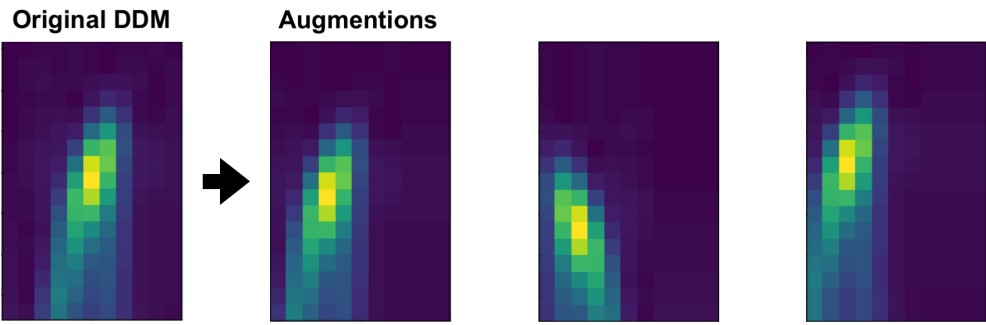

**Figure 8.** Example of randomized image augmentation (translations/reflections) applied to a DDM (horizontal and vertical translations are exaggerated for illustrative purposes).

Due to the vast number of tunable hyperparameters with any network, and seeing how that number increases with the addition of new regularization strategies, layer types, optimizers, etc., this study is not meant to assert the optimal architecture to use in the context of DDMs, but rather a comparison of strategies evaluated when informing the network architecture selected for this study. However, in seeing consistently strong performance from the CNNs, we believe that there is justification for their application in the analysis of DDMs. If future missions provide reflections measurements with higher resolution DDMs, the advantages of CNNs may become even greater when applied to these richer datasets [39,40].

### 2.2.5. Complete Soil Moisture Network

In parallel with the CNN development, a complete neural network for the prediction of soil moisture was developed using a simplified CNN as a placeholder. The various layers of this network were constructed and adjusted through common heuristics and experimentation to yield reasonable performance. As depicted in Figure 6, the final network consisted of 3 major components: the convolutional layers described in the previous subsection (blue), the following concatenation layers (yellow), and the densely connected layers (pink).

The concatenation layers performed the task of combining elements from the various ancillary inputs with the flattened output of the convolutional layers. The software defining the network's architecture was designed to construct a layer with variable ancillary inputs, allowing for flexibility to experiment with different input combinations. As detailed in Section 2.1, these inputs could be either ordinal or categorical in nature. Here, ordinal means that the value of the input corresponds to some kind of continuous, ordered parameter, such as an angle or surface-water fraction, while categorical inputs have values that do not necessarily correspond to one another (spacecraft ID numbers, or arbitrary land type indices). The former are input as single values, while the latter are encoded into "one-hot" arrays (discussed in Section 2.1.3). This resulted in a dynamically sized initial concatenation layer which varied with the particular choice of inputs. In future work, we intend to explore categorical embeddings as a replacement for one-hot encodings, as the former has shown benefits in various learning applications across the deep-learning community.

The dense layers of the network were composed of 4 fully connected layers after the initial input layer from concatenation, fixed in dimensions of 128, 64, 64, 64. The ReLU

activation function was used due to initial comparison of training speed and performance, as well as its common appearance in most modern architectures. Future experimentation may include a change from ReLU to SWISH, an activation function similar in characteristic to ReLU but demonstrated to match and often exceed the performance of ReLU when replaced in a series of benchmark networks [41]. The final layer combined to a single valued output, which yields the prediction of soil moisture from the network.

In the final stages of the work, the DDM-tuned CNN was swapped into the network. Due to the inherent complexity in developing a neural network architecture to meet the needs of multiple input parameter types and constrained by time and budget, after finding relative success with the selection of dense layer hyperparameters, the team decided to keep those layers fixed in favor of focusing efforts on areas perceived to have more benefit.

The network was trained in a sequence of "epochs" using mini-batch gradient descent, where the training data were randomly split into subsets (batches) and the network trains, updating its weights with each pass of a batch. At the end of the training epoch, once all batches had been utilized, the model was fed the validation dataset, and the performance was evaluated. New randomly sampled batches were created for the next epoch, and the process repeated. This was performed 36 times per training, a value where the losses appear to roughly asymptote. During this process, the training/validation loss curves were consistently evaluated to recognize signs of the network overfitting to the training data. Mean squared error (MSE) was selected as the loss function due to its use by other soil moisture-data products for quality measurement.

## 3. Results

### 3.1. Performance Analysis

Performance of the network's predictive capabilities was evaluated using two metrics: minimization of the overall error between the target and predicted values, and reproduction of the target variable distribution function. These are different goals in the sense that a small residual from our loss function could be attained by good prediction of the majority of samples, while ignoring less common, but important, fractions of the soil moisture distribution. This could also relate to poor regional performance, geographically. In follow-on work, a custom loss function integrating a measure such as the earth mover's distance (EMD) could help the network learn to better reproduce the target distribution, an approach that may remove the need for over/undersampling the training data. Considering this, we used several analysis techniques to measure the output of the network during the development process.

Performance was established using the test-data split from the original dataset. As detailed in Section 2.1.8, the test set is composed of randomly selected samples from the original dataset, prior to balancing. This yields a metric of comparison representative of the actual measured soil moisture distribution. As such, the training-set balancing process can result in slight variability in agreement between the target and prediction distribution for very high soil moisture values. This depends on which high soil moisture samples are split into the test set versus samples that are highly oversampled in the training set. Figures 9–11 represent an example analysis of a typical training output.

By passing these "new" test samples through the network, the correlation of the target and prediction forms our first metric of performance. Figure 9, top, shows an example correlation between prediction and test target, with the orange line indicating ideal correlation. Here, color represents counts on a logarithmic scale indicating the majority of predictions values fall near the orange line, though there are outliers that deviate from the targets substantially. In addition, note the sharp boundary in predictions at around 0.55 $m^3/m^3$, indicating the edge of the balanced training-set distribution. A Pearson coefficient of 0.89 for this typical run indicates strong correlation between the predictions and targets.

Observation of the differences as a function of soil moisture are more easily seen by overplotting the distributions, as shown in Figure 9, bottom, where the target distribution

is shown in blue, and prediction in orange. We see that, while representation of the target distribution is not perfect, the overall nature of the distribution is captured. Values of high soil moisture are also represented in the predictions, a capability enabled by the previously described sample-balancing process. In addition, note the sharp cutoff of the target distribution at 0.02 $m^3/m^3$ is not captured.

Analyses of the form shown in Figure 9 also enable detection of artificial or unphysical behavior. An example of this is the spike in the prediction distribution near 0.28 $m^3/m^3$, which appears as a horizontal band in the correlation plot. Predictive behavior such as this indicates a possible issue in the training process where certain soil moisture values are predicted regardless of the input. This could result from network architectural issues or problems in preprocessing steps. Similar behavior in the target distribution (vertical lines in the correlation plot) can occur when there are certain target values that contain unreliable measurements. Examples such as these demonstrate the benefit of an iterative loop between preprocessing, training, and analysis.

We also looked at the difference between target and prediction, or the error. Figure 10 shows the absolute error (top) and the percent error (normalized by target, bottom) for the same data shown in Figure 9. We see a bias towards underestimation of soil moisture for large target values, but a narrowing of the actual error for these values. This bias may be partially a result of the balancing process discussed in previous sections. The significant spread in percent error for dry targets is due to small errors being large relative to the target, but there is a clear bias toward overestimation of the wetness in dry regions. Note that this is partially due to the artificial cut off in the SMAP values at 0.02 $m^3/m^3$, which is not captured by the predictions. In rare cases the predictions can also be negative, which is clearly unphysical, and will be constrained in future work.

The analyses above focus on the study of the predictive capability with respect to target distribution, but this ignores the nontrivial geographic effects. Figure 11 shows the spatial distribution of the error (top) and percent error (bottom), giving us a clearer picture of where deviations in predictive capability are occurring. Significant relative overestimation occurs over the planet's driest region, the Sahara, while underestimation is common in the Amazon, the Congo, and coastal areas. In addition, note the missing regions from the map, including most of the track of the Amazon where many unreliable SMAP values occur, as well as the Tibetan Plateau and the Andes, which are removed due to surface elevation. These are parts of the planet that the network is never directly exposed to in the training process, but can provide estimates when fed the validation samples; a strategy for evaluating the quality of these estimates has yet to be developed.

### 3.2. Global Product Comparison

Through trial and error, a combination of ancillary inputs and filtering was found which yielded predictions with reasonable correlation to their targets for a substantial fraction of the regions covered by CYGNSS. The ancillary sets used include those detailed in Section 2.1.1, and filtering used the limits outlined in Section 2.1.7. These choices are not necessarily optimal, but were sufficient for this proof-of-principle study. A more extensive analysis will be part of future work to develop the required tools for determination of optimized parameters. This configuration was used to generate a soil moisture product from CYGNSS DDMs and ancillary data for the validation window.

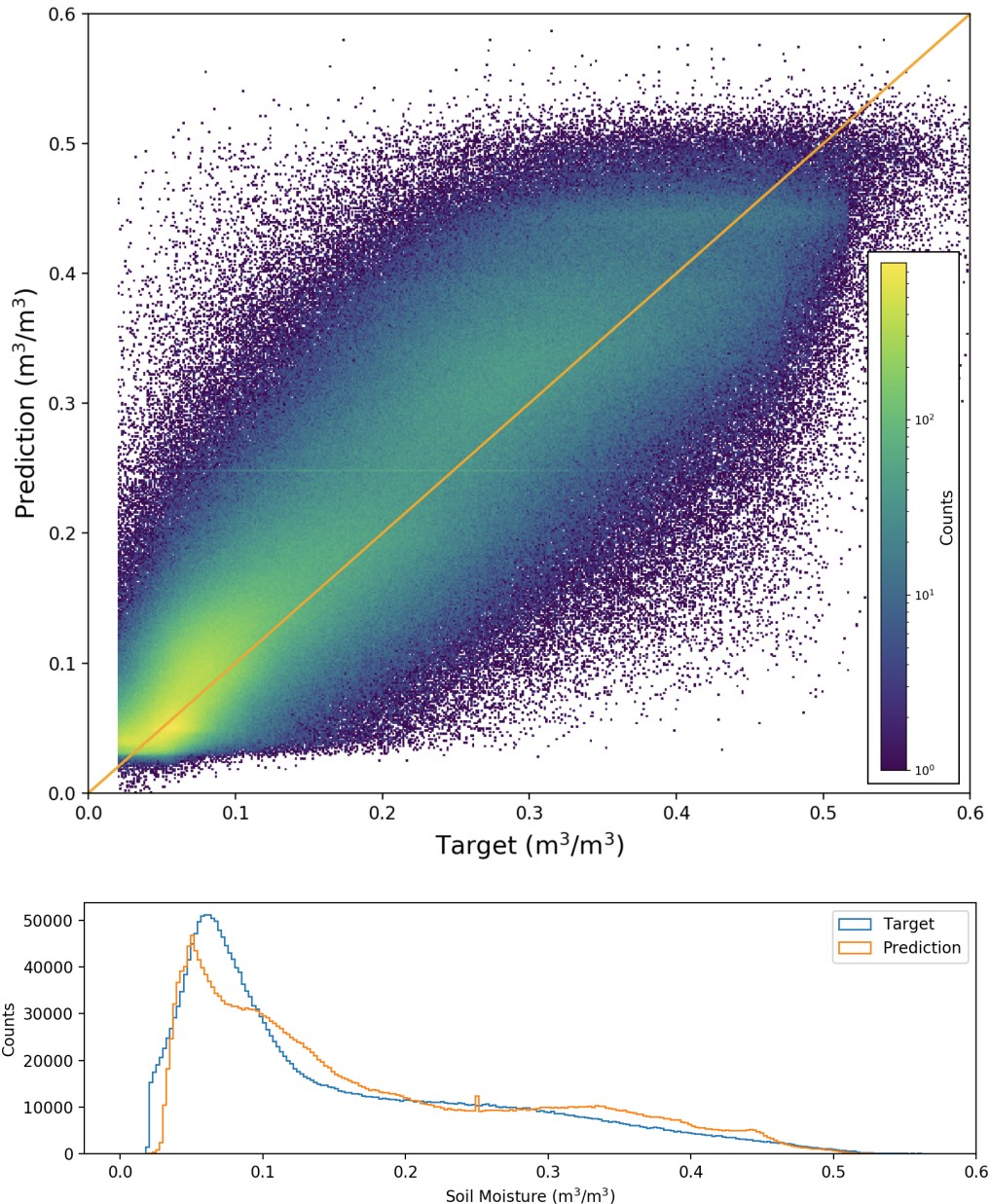

**Figure 9.** (**Top**) Results from an example training session showing the correlation between prediction/target where color is on a logarithmic scale. Orange line highlights ideal correlation. (**Bottom**) Comparison of prediction and target distributions.

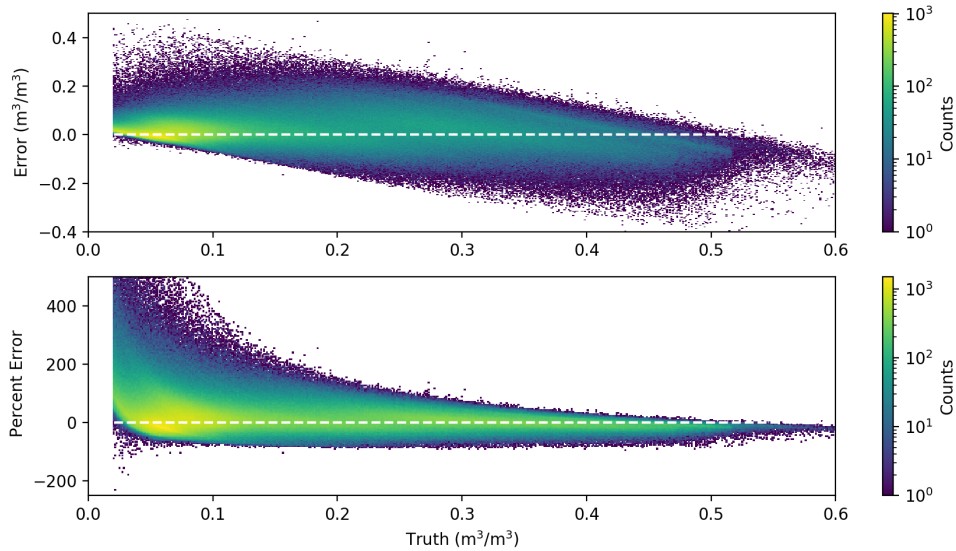

**Figure 10.** Error (**top**) and percent error (**bottom**) between target and prediction as function of target value for same run as shown in Figure 9.

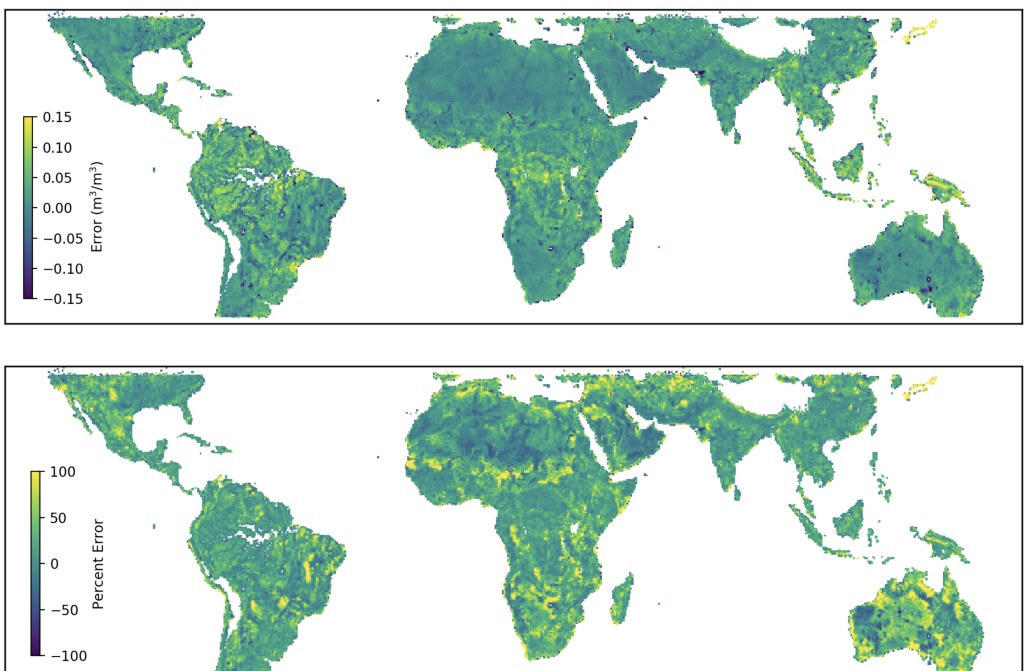

**Figure 11.** Error (**top**) and percent error (**bottom**) between target and prediction for same run as shown in Figure 9.

A goal of this work was a direct comparison to mature CYGNSS-based soil moisture products. As mentioned, the product from UCAR developed by Chew et al. [26] uses a method which forms a linear relation between the peak power of the DDM (corrected using Equation (1)) to soil moisture as measured by SMAP, an example of more traditional DDM interpretation which ultimately utilizes only a single value per DDM. This is performed through a calibration scheme which creates a global look-up table of these relations on a 36 km EASE grid. This technique has generated a valuable soil moisture product, but has potential to underperform in regions where the relation between peak power and soil moisture is conflated with other variables.

We performed a global comparison of the network predictions to the SMAP measurements and the product developed by UCAR by averaging values from late 2018–2019 to 36 km EASE grid cells. Figure 12 shows these averages for SMAP (top), UCAR (middle) and CNN (bottom). Qualitative comparison of these figures shows the similarity in the overall global trends. However, areas with exceedingly high soil moisture content as measured by SMAP (particularly the Amazon, the Congo, and some coastal regions) display a less detailed structure, as is highlighted in Figure 13. As will be discussed in the following section, there is a correlation between extremely high soil moisture estimates from SMAP and UCAR, such as the Amazon river's tributaries, and SMAP flags indicating low-quality measurements.

The global unbiased root mean square error (ubRMSE) is calculated for each cell by removing the validation set averaged mean value (bias) from the daily CNN predictions, performing the same for the SMAP measurement, and calculating the root mean square error in time. The values shown in Figure 14 represent the validation-period averaged ubRMSE.

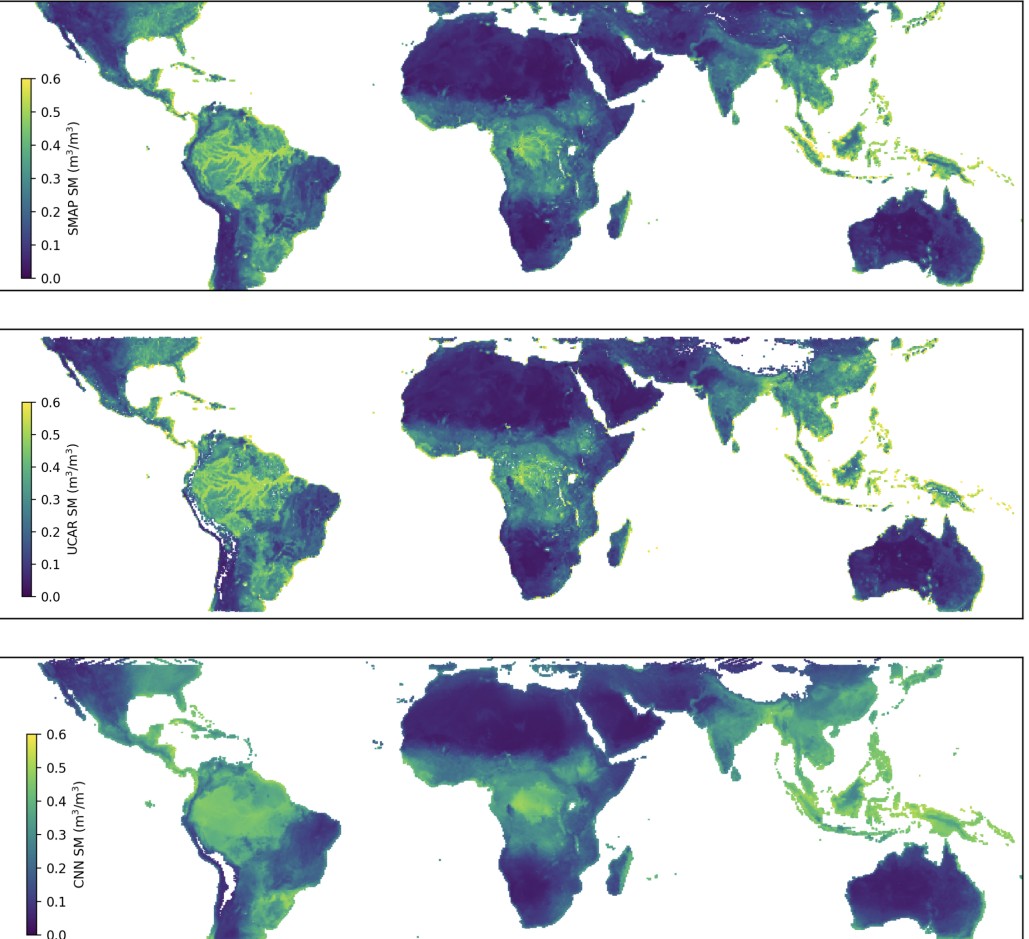

**Figure 12.** Global soil moisture from SMAP (**top**), UCAR (**middle**), and the neural network (**bottom**), averaged over late 2018–2019 to 36 km grid cells, shows similar large-scale trends.

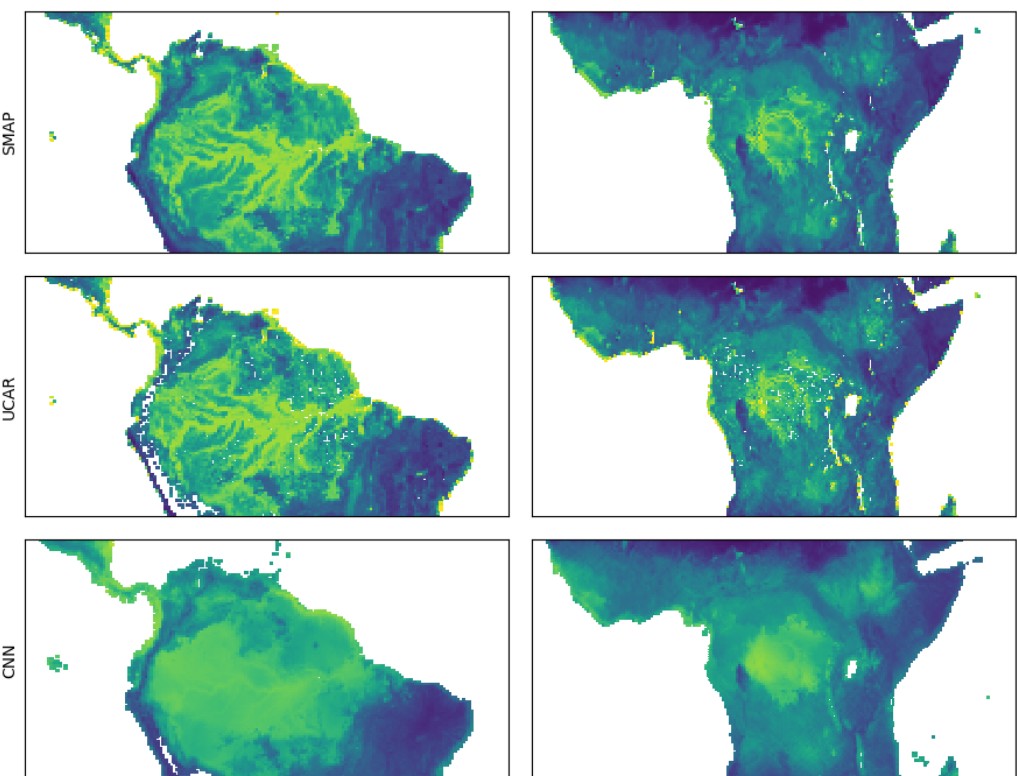

**Figure 13.** Comparison of the Amazon region (**left**) and the Congo region (**right**) between SMAP (**top**), UCAR (**middle**), and CNN (**bottom**). Regions with high soil moisture (yellow) in upper two plots are associated with low-quality SMAP values, which were excluded from the training process.

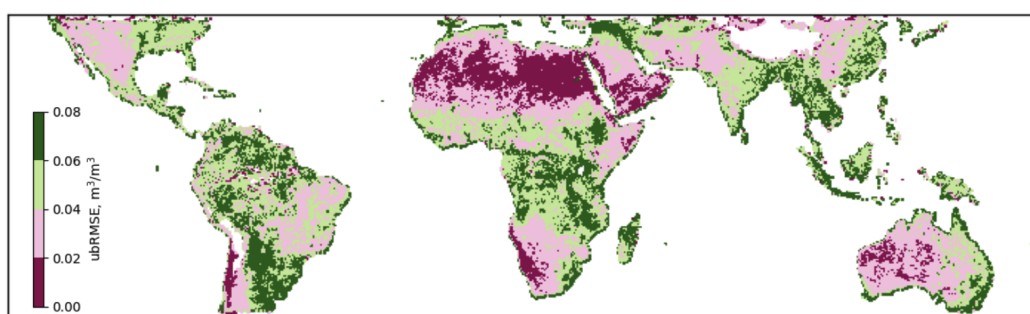

**Figure 14.** Global ubRMSE between the daily soil moisture predictions of the CNN and SMAP. Note regions lacking measurements from either dataset appear white.

Comparison as a function of soil moisture is made in Figure 15, which overlays the 36 km averaged daily soil moisture distributions of the three products. Here, we see ranges of values where the CNN and UCAR agree, and other ranges where there is clear disagreement. Note that the peaks in soil moisture measurements near 0.5 m$^3$/m$^3$ in the UCAR and SMAP distributions are related to SMAP measurements with quality issues, which the CNN dataset does not reproduce.

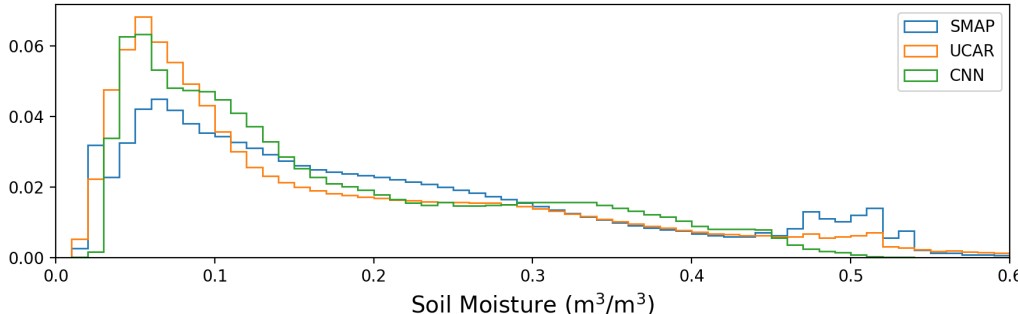

**Figure 15.** Comparison of the SMAP, UCAR, and CNN-based soil moisture distribution from the validation window. The features near 0.5 m$^3$/m$^3$ in the SMAP/UCAR distributions are related to low-quality flags in regions such as the Amazon.

The above comparisons were performed at the lowest common spatial resolution between the three data products, 36 km, set by the binning process UCAR uses to establish local relations between SMAP and CYGNSS measurements. The training process for the CNN does not have this requirement, and, therefore, the measurements can be resolved to finer spatial scales. The ancillary data were spatially averaged to 3 km resolution grids to match the average footprint of the DDM's peak pixel (an approximation of the coherent reflection region), setting a rough lower limit for the spatial resolution of GNSS-R. Figure 16 compares the validation-window averaged product at 36 km (top) to the same values binned to 3 km bins, showing the attainable resolution with this technique.

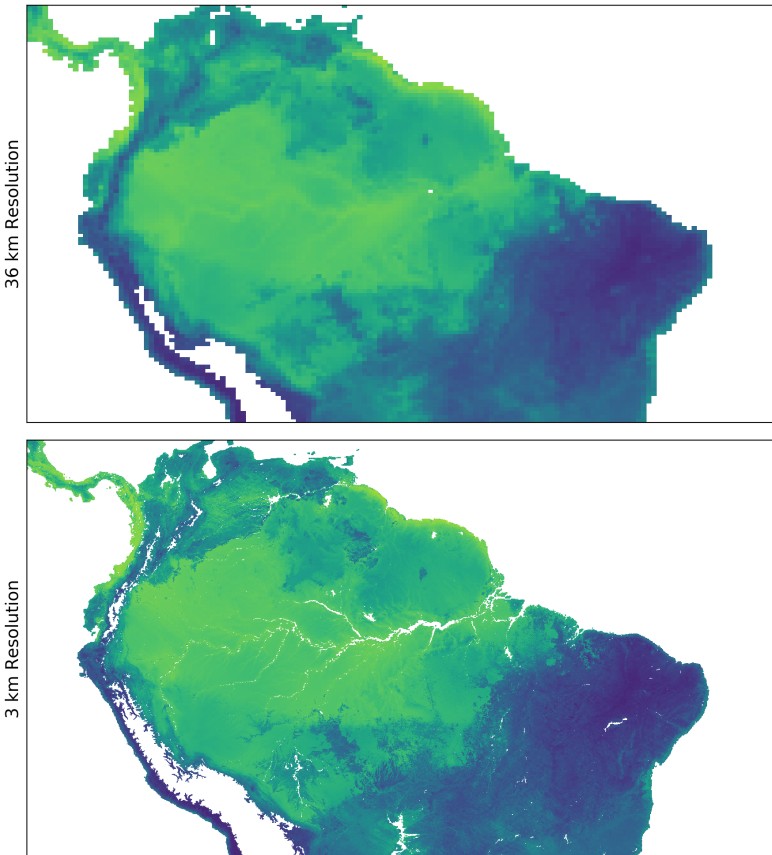

**Figure 16.** Comparison of the 36 km resolution (matching UCAR) to the 3 km resolution achievable with the CNN-based approach. White regions are those filtered due to surface water or topography considerations.

### 3.3. In-Situ Product Comparison

Ground stations from the USCRN soil moisture network [42,43] were used as validation of the CNN predictions, and allow for direct comparison to the performance of other remote-sensing-based estimates. Stations were selected across the continental United States (the 28 station locations shown in Figure 17) wherever there was sufficient data for analysis from SMAP, CYGNSS, and USCRN. Figure 18 shows the soil moisture measurements (5 cm depth) from six locations in the southern United States as solid lines, and the predictions from SMAP (blue), UCAR (orange) and the CNN (green) as points. Agreement in trends of these products with the in-situ measurements was quantified through calculation of the Pearson correlation coefficient for each site, as shown in the upper plot of Figure 19. Differences in performance across sites and products are observed. Additionally, Figure 20 shows the ubRMSE calculated (as described above) for all three remotely sensed products (SMAP, UCAR, CNN) at each station, averaged across the validation window.

This analysis also provides a direct method to observe the performance of the CNN under different input configurations. A limited scope input ablation test was performed to study the ability of the CNN to make predictions at these sites. The four input conditions were (1) using only the 2D DDM array, scaled to effective surface reflectivity (Equation (1)); (2) using only ancillary data (no input from the DDM); (3) using only the peak of the (scaled) DDM, with the full ancillary input; and (4) using the full input dataset. The lower plot of Figure 19 shows the correlation of the CNN predictions for these four sets of network configurations.

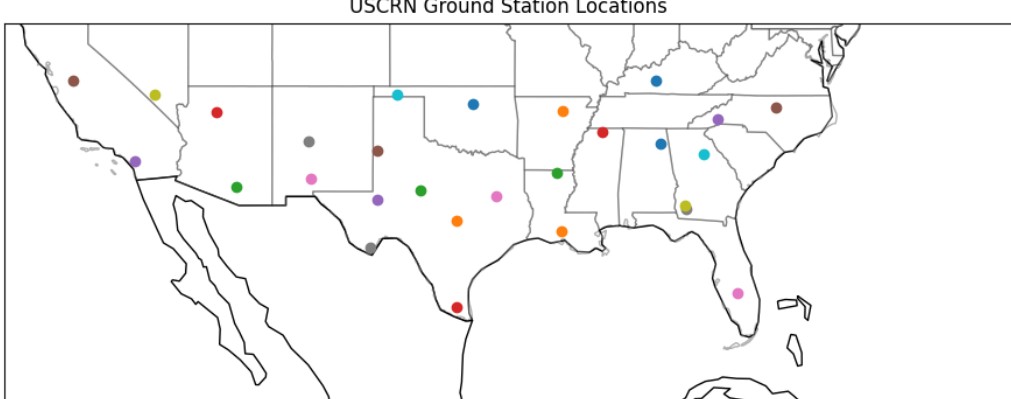

**Figure 17.** Locations of the 28 USCRN ground stations discussed. Stations were selected to provide geographic coverage from a consistent source.

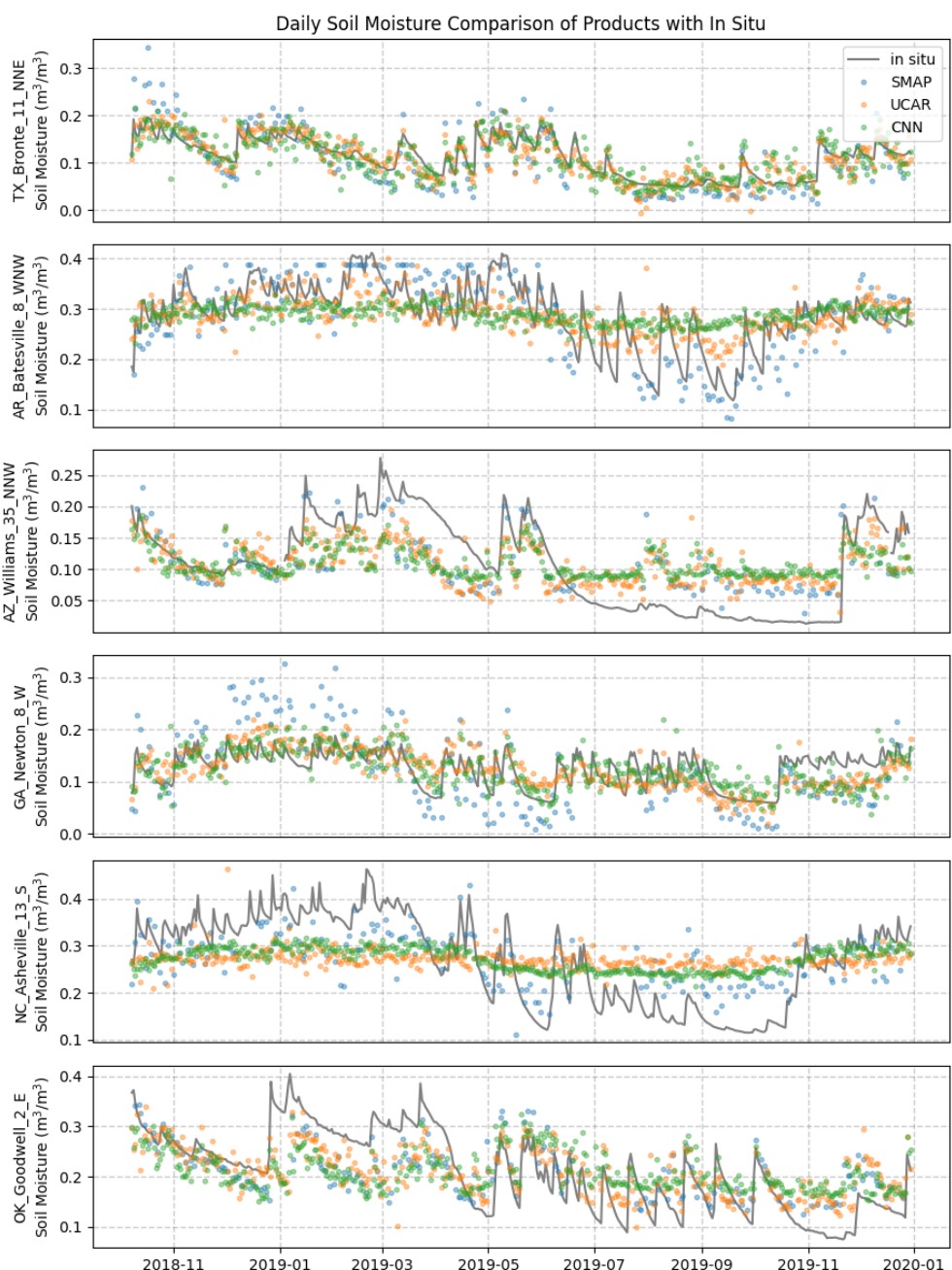

**Figure 18.** Comparison of the soil moisture measurements from six locations in the southern United States (solid lines), and the predictions from SMAP (blue), UCAR (orange), and the CNN (green), from daily averages 36 km around the site. In-situ values are from the USCRN 5 cm depth soil moisture product. Note that the bias between each product and the in-situ measurement was subtracted.

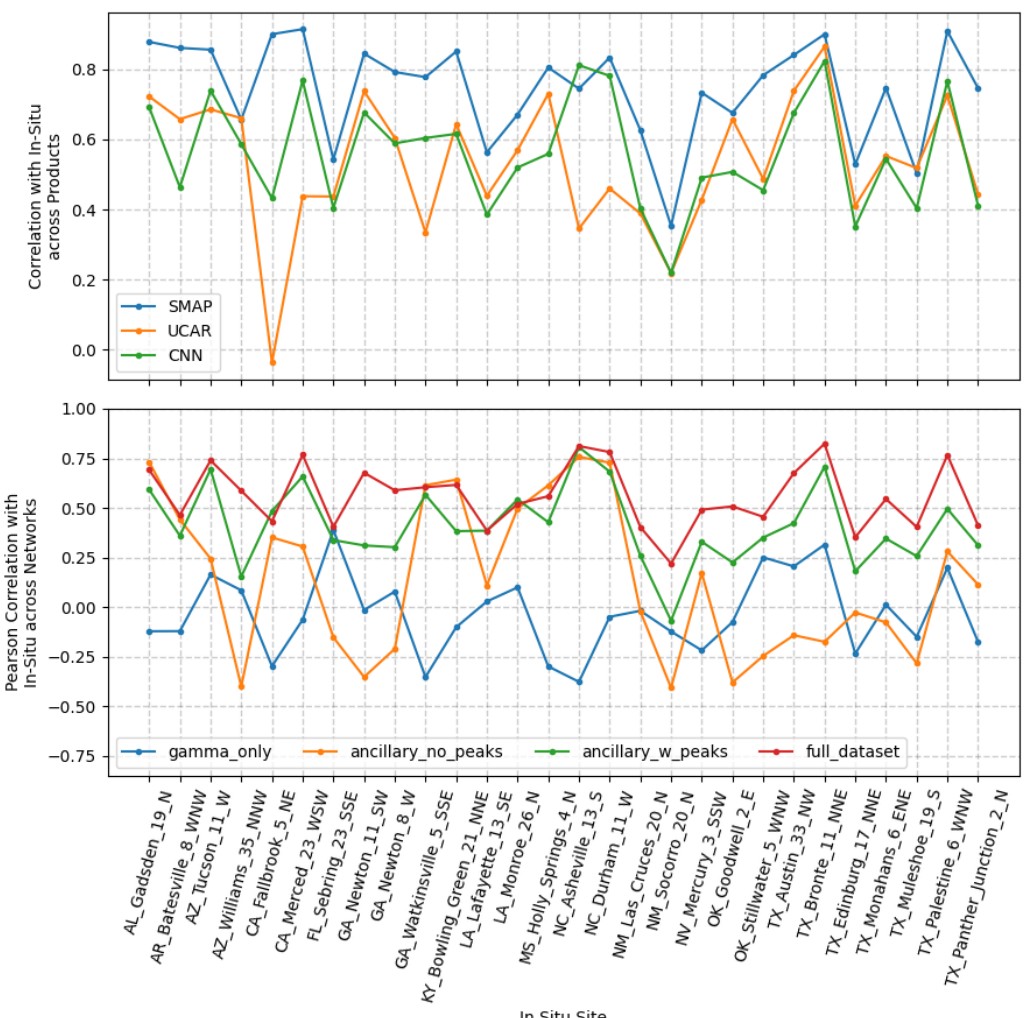

**Figure 19.** Pearson correlation coefficients were used to quantify the performance of the products against in-situ measurements of soil moisture of 5 cm depth. Upper, the correlation of the three remote products (SMAP, UCAR, CNN) against the measurements from 28 ground stations. Lower, the correlation of the CNN predictions for various input configurations showing the general improvement in the network performance with increasing information.

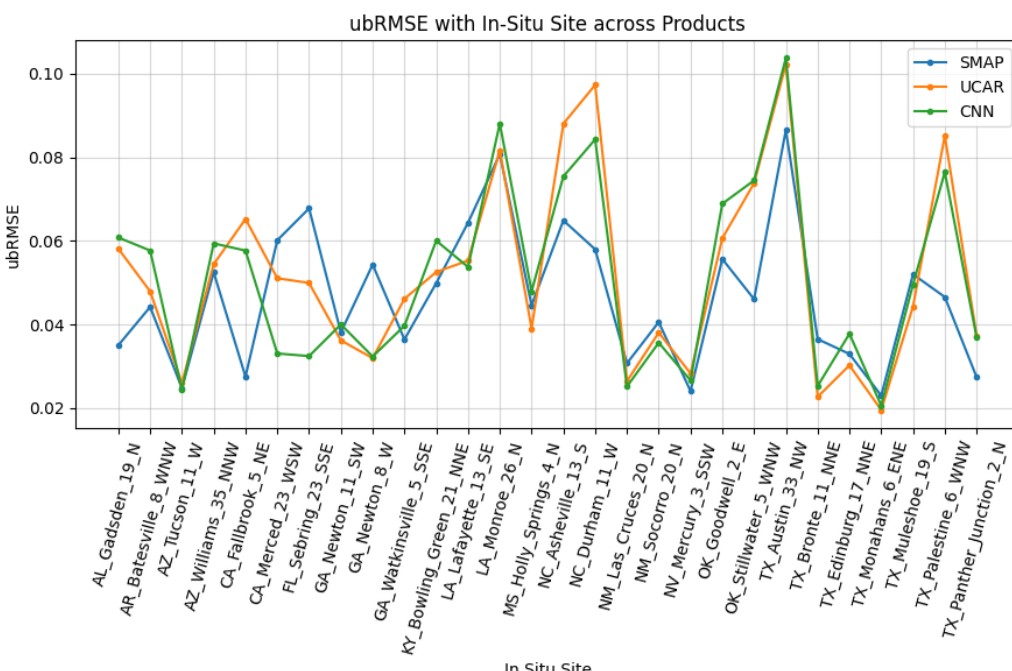

**Figure 20.** Average ubRMSE for the validation window calculated for each of the three remotely sensed products at the USCRN stations.

## 4. Discussion

The results shown in the previous two sections demonstrate the predictive capability of using a convolutional neural network to ingest and interpret 2D delay-Doppler maps. While preliminary, these findings show that a CNN can create soil moisture predictions that correlate well with SMAP measurements, producing estimates with a generally comparable distribution, both geographically and as a function of soil moisture. We now consider, in more detail, several of the more interesting features from these results.

The disagreement in regions of high soil moisture raises the question of whether these very high measurements reported by SMAP are invalid, or whether this is a manifestation of limits imposed by the training process. The soil moisture in a volume is fundamentally limited by the amount of pore space in the soil. When the pore space is filled, the soil is saturated and water will pool on the surface until cleared by other mechanisms. Therefore, the porosity of soil, or the ratio of non-solid volume to the total volume, sets an effective upper bound on soil moisture [44]. Figure 21 shows the porosity for the 0–30 cm soil layer, indicating that the volumetric limit for soil moisture is closer to 0.5 $m^3/m^3$, but is generally lower than this in regions that were reporting the highest values from SMAP and UCAR (e.g., the Amazon). These values, which are likely incorrect, could be driven by the presence of dense vegetation, which affects the emissions collected by SMAP [30,31].

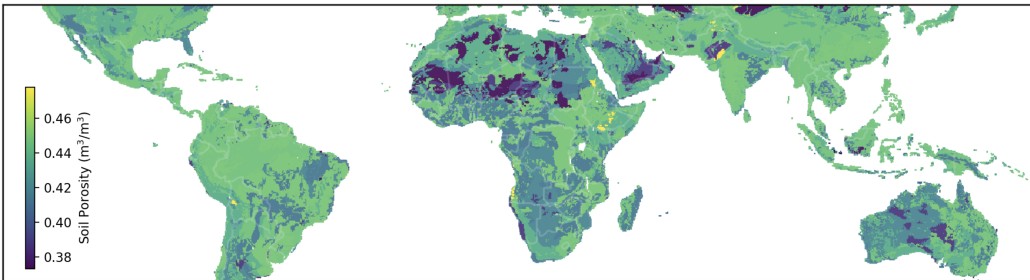

**Figure 21.** Map of soil porosity for measurement region indicating effective upper bound on soil moisture value.

Accordingly, many of the SMAP values reported in this region are marked by flags indicating they are of low quality. The upper panel of Figure 22 shows the majority of soil moisture values from the South American region, which are associated with the red distribution of Figure 4. Note that the regions tracing the path of the Amazon River are explicitly missing from this map. Compare this to the lower panel of Figure 22, which shows the SMAP values that were excluded from the training dataset (purple and black in Figure 4), which appear to contain all of the exceedingly high soil moisture measurements in the region. Exclusion of these samples from the training process was performed in part due to the irregular distribution of the samples (as seen in Figure 4), but also due to the appearance of poorly correlated predictions in our assessment of network output, an indication of problematic input values. This effect likely explains the majority of the discrepancies between our predictions and those of SMAP and UCAR over regions with high soil moisture.

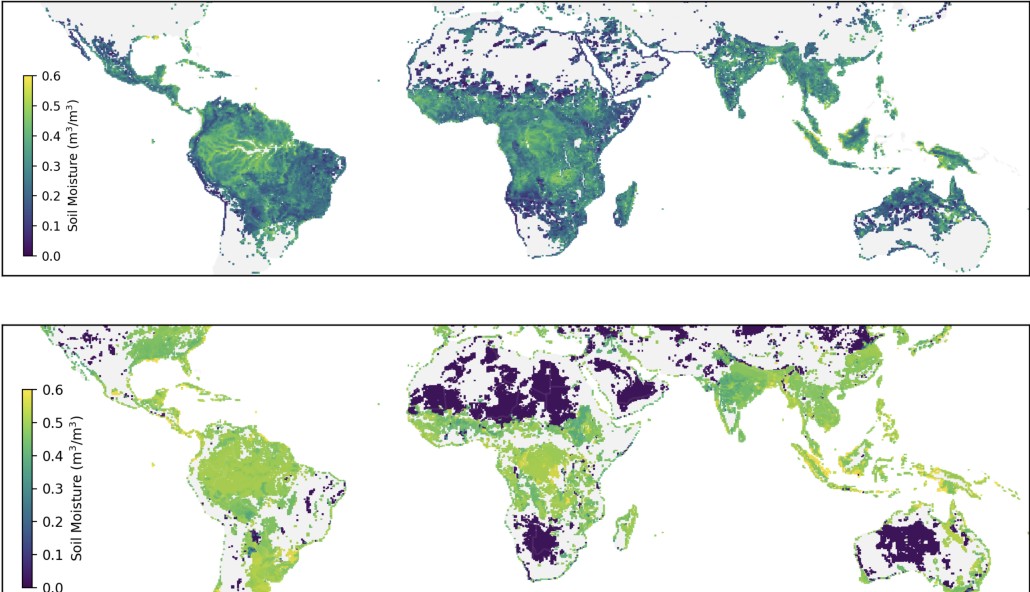

**Figure 22.** SMAP soil moisture values associated with two classes of "low-quality" flags, the upper panel were included in training, the lower panel were not.

As discussed in previous sections, the effects of training set distribution balancing are beneficial, but can lead to artifacts in the predictive behavior of the network. The clear boundary in prediction values at $0.55 \, \text{m}^3/\text{m}^3$ visible in Figure 9 is one marker of this. Other choices in filtering could also contribute to the disagreement, such as biases created by removing samples from regions with high surface water fraction. While data preprocessing is clearly a possible driver, these effects are likely secondary to the differences caused by problematic SMAP values. Understanding whether these differences are acceptable, and/or improving the preprocessing to further avoid bias, will be a key focus of continued work. An alternate approach that may better resolve this issue could utilize multiple neural networks in a tiered manner. An initial network would operate as a binary classifier, labeling samples in one of two categories: "low soil moisture" and "high soil moisture". Two additional networks would be trained to estimate soil moisture from the "low" and "high" regions of the soil moisture distribution. Samples would receive an initial "low"/"high" classification that would determine which secondary network to estimate the sample's soil moisture value. Alternative loss functions, such as EMD, will also be explored.

An advantage to the CNN-based approach presented here over that of UCAR's product is that regions where problematic SMAP values were excluded (such as over the Amazon) are still able to be evaluated by the network for reflections that occur there. This is not the case for UCAR's product, where the look-up table is inherently geographically bound.

For a given location, a sufficient number of SMAP values must be aggregated to build a relation with the DDM peak values. This necessitates the inclusion of problematic SMAP values over the Amazon. On the other hand, with a CNN, latitude and longitude were simply two of many supplementary inputs. If the network was never exposed to a specific location, information is still provided by the rough localization to the region as well as other contextual information. This allows the network to make a "best guess" at a previously unseen location, helping to guide the estimation of soil moisture from the DDM. Note that the UCAR product contains propagated quality flags that allow for the exclusion of these bad regions, yet this will result in an area with no predictions.

We anticipate that having error estimates associated with predictions made on unseen regions may be useful. Generating these error estimates could be achieved by removing well-sampled regions (or strips) from the training, validation, and test splits. After retraining the network and ensuring reasonable performance, samples from the removed regions would be fed through the network and errors calculated. These errors would serve as a proxy for estimating errors in regions such as the Amazon.

It should be pointed out that the network predictions are inherently limited to the accuracy of SMAP's measurements. With SMAP's ubRMSE of 0.04 m$^3$/m$^3$ [36], it is reasonable to expect high relative error over the driest regions of the planet where the soil moisture might be on the order of this uncertainty. This can explain the regions of high percent error over the Sahara, for example, shown in Figure 11. As mentioned in Section 2.1.5, spatially and temporally varying biases of SMAP's measurements relative to in-situ ground stations [36] necessitate an offset removal for accurate comparison. These same biases are subsequently passed to the predictions of the network during training. Therefore, the predictions should be thought of as "predictions of SMAP values" as opposed to predictions of explicit soil moisture values, as would be measured by any specific ground station. This is a general challenge related to possible instrument calibration issues and/or highly localized sampling by in-situ measurements.

The comparison of spatial resolution in Figure 16 displays the capability for high-resolution soil moisture estimation attainable with this process. This comes with the inherent trade off in coverage at these finer resolutions, which necessitates reduced temporal resolution to accommodate. Additionally, study of Figure 16 shows indicators of the CYGNSS tracks, an unphysical feature in soil moisture. These, and other details, will be studied and refined in future work.

Comparisons to in-situ measurements of soil moisture from the USCRN network of ground stations, shown in Figures 18 and 19, indicate a strong correlation with the remotely sensed estimates. In particular, peaks from precipitation events and many drying/drainage features are well captured at sites Bronte (TX), William (AZ), and Goodwell (OK). Other sites, such as Batesville (AR) and Ashville (NC), display a similar lack of correlation as seen in the SMAP/UCAR results. This said, there are several points in Figure 19 where the CNN appears to outperform the UCAR product's ability to observe certain trends. This could be a result of information captured in the ancillary datasets, such as seasonally varying precipitation, not available to the UCAR analysis. The ubRMSE at these sites (shown in Figure 20) produces examples where each product outperforms or underperforms all others, but, overall, SMAP appears to have the lowest variation while the trends between UCAR and the CNN are, unsurprisingly, more similar.

The input ablation study mentioned in Section 3.1 shows some interesting, yet expected, behaviors in network performance. We see limited predictive capability of the network when only DDM information is used, or when only ancillary information is used, indicating that the combination of these sources is required to reasonably estimate soil moisture. The third case, using the ancillary data but only the peak of the DDM power (scaled to the effective surface reflectivity), is an approach without the use of the full 2D input. This is a somewhat analogous, albeit limited, approach to that performed by others [23,24], and represents an intermediate level of data ingestion, yielding good performance, but not as good as the final case where the full DDM is ingested. In this final input configuration,

we see the best performance, indicating that the inclusion of information from the 2D DDM is important to improving the accuracy of SM estimation.

This project has demonstrated a new technique for measuring soil moisture at greater spatiotemporal resolution than is currently provided by SMAP by leveraging the spatial structures and information contained in the full 2D DDM, thereby enhancing these valuable measurements. This exercise has built experience with the related datasets, and refined our understanding of the network architectures useful for the process, setting the stage for a follow on, complete soil moisture analysis and, more generally, validated the concept of processing DDMs with CNNs. The code for performing this analysis, end-to-end, as well as scripts to download the related datasets will be provided for community use in the near future.

## 5. Conclusions

Low-cost GNSS-R constellations such as CYGNSS can estimate surface parameters with high spatiotemporal resolution relative to single-spacecraft missions such as SMAP. This comes at the expense of requiring a calibration process to somehow relate the GNSS-R measurements to a "known" surface value. We have successfully demonstrated that a CNN-based analysis of the 2D DDM can be used to estimate soil moisture when trained on SMAP data. Inclusion of the 2D DDM provides a more complete picture of the reflection, yielding the potential for this approach to have advantages over reported ANN methods using only features of each DDM as input [22–24]. Due to the simplicity of integrating additional contextual data, there is a potential for this process to yield improved accuracy and coverage compared to existing techniques for soil moisture using CYGNSS data. While the work described here only used spatially averaged ancillary data, it is imaginable to process arrays of local information as ancillary inputs, say, a 2D array of terrain slopes, or vegetation parameters. This concept would be challenging to model analytically for use with traditional forms of analysis, but requires only a simple modification of the network architecture to implement. Furthermore, this study has laid the groundwork for exploring how GNSS-R can be used for other surface retrievals beyond soil moisture. A slight modification in this pipeline would allow for the predication of freeze/thaw conditions also trainable with SMAP data. The parallel work in land type classification has shown the potential for estimation of this high-level surface parameter. Flooding, inundation, and water mask development would be prime targets for prediction as well. In addition to extending the retrieval possibilities in ground-based analysis, these machine-learning-based techniques have potential in instrument-level processing applications, including adaptive targeting, and "smart" data prioritization through classification for downlink constrained systems.

**Author Contributions:** The conceptualization, methodology, resources, writing, review and editing of this work was performed by all authors. The data curation, development of software, formal analysis, visualization, validation, and funding acquisition, were primarily performed by T.M.R. and I.C. All authors have read and agreed to the published version of the manuscript.

**Funding:** The work described in this paper was carried out by the Jet Propulsion Laboratory, California Institute of Technology under a contract with the National Aeronautics and Space Administration (80NM0018D0004). A portion of this work was funded by the University of Michigan as part of the CYGNSS Extended Mission Support.

**Data Availability Statement:** Not applicable.

**Acknowledgments:** The GEOS data used in this study/project were provided by the Global Modeling and Assimilation Office (GMAO) at NASA Goddard Space Flight Center.

**Conflicts of Interest:** The authors declare no conflict of interest.

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
