# Peer review of "A Deep-Learning Approach to Soil Moisture Estimation with GNSS-R"

_remotesensing, doi:10.3390/rs14143299_

Round 1
Reviewer 1 Report
Basically, the paper deals with the demonstration that a convolutional neural network trained with the 2D Delay-Doppler Maps from the Cyclone Global Navigation Satellite System constellation and contextual ancillary datasets, such as surface topography and water, normalized difference vegetation index, VWC, and land coverage, and aligned with SMAP soil moisture values as the targets can be used to generate global soil moisture products with high temporal and spatial resolution.
The manuscript is well written, is easy to follow, and the conclusions are based on the results and their analysis.
However, some minor changes have to be introduced:
- Line 231. Extra explanations about the definition of unbiased RMS error (ubRMSE) are needed.
- Line 281-282. What do you mean by removal of spatially and temporally aligned samples in the complete database? It is related with high correlation samples?
- Section 2.1.5. How do you build the balance representation for the training model in high soil moisture values, by repetition of those that exist?
- Lines 488-489. Why do you think a change from ReLU to SWISH in needed?
Reviewer 2 Report
The article proposes a CNN-based approach to estimate soil moisture from CYGNSS data. Despite the interest of this development, the article requires significant improvements before an acceptance.
1) The introduction needs a rewrite, the state of art of soil moisture estimation by GNSS-R data is not enough described (more references), at the beginning of the introduction, it would be useful to better introduce the need for have soil moisture….
2) The database, the auxiliary data are not enough described (sources of NDVI data, etc.) + field data used during validations.
3) The method section is very difficult to follow, a very strong need for a better structuring of the work carried out and the various methods and tests. There is too much information being mixed up.
4) We do not see enough the contribution of the proposed approach compared to works based on a simple ANN between CYGNSS data and soil moisture, to be deepened?
5) The choice of field data: why not take a more complex context than that of the United States (ISMN network).
6) Is it possible to add an analysis of contribution of each input variable in soil moisture precision ?
7) why the use of just two years of data ?
Round 2
Reviewer 2 Report
Authors answer different comments. I propose acceptance of the paper.